# Combining Child Functioning Data with Learning and Support Needs Data to Create Disability-Identification Algorithms in Fiji’s Education Management Information System

**DOI:** 10.3390/ijerph18179413

**Published:** 2021-09-06

**Authors:** Beth Sprunt, Manjula Marella

**Affiliations:** Nossal Institute for Global Health, Melbourne School of Population and Global Health, The University of Melbourne, Melbourne, VIC 3000, Australia; beth.sprunt@unimelb.edu.au

**Keywords:** child functioning module, learning-support needs, disability-inclusive education, Education Management Information System, Fiji

## Abstract

Disability disaggregation of Fiji’s Education Management Information System (FEMIS) is required to determine eligibility for inclusive education grants. Data from the UNICEF/Washington Group Child Functioning Module (CFM) alone is not accurate enough to identify disabilities for this purpose. This study explores whether combining activity and participation data from the CFM with data on environmental factors specific to learning and support needs (LSN) more accurately identifies children with disabilities. A survey on questions related to children’s LSN (personal assistance, adaptations to learning, or assessment and assistive technology) was administered to teachers within a broader diagnostic accuracy study. Descriptive statistics and correlations were used to analyze relationships between functioning and LSN. While CFM data are useful in distinguishing between disability domains, LSN data are useful in strengthening the accuracy of disability severity data and, crucially, in identifying which children have disability amongst those reported as having some difficulty on the CFM. Combining activity and participation data from the CFM with environmental factors data through algorithms may increase the accuracy of domain-specific disability identification. Amongst children reported as having some difficulty on the CFM, those with disabilities are effectively identified through the addition of LSN data.

## 1. Introduction

To provide quality education for all, education systems in low- and middle-income countries are striving to transform to meet the diverse learning needs of all children. This requires approaches and policies that understand and value diversity in students’ abilities, needs, and individual characteristics. Whilst access to disability-inclusive education should ideally be accorded without requirements for eligibility, in reality, providing special measures, such as reasonable accommodation or individual supports, means that definitions and parameters for eligibility must be determined [1].

Good evidence regarding the situation and needs of students with disabilities is a critical element in quality education for all. The central data mechanism within ministries of education that enables this evidence for policy development, planning, and budgetary allocation is the Education Management Information System (EMIS). For this system to support disability-inclusive education, student data must be disaggregated by disability. Methods of disaggregating EMISs by disability are evolving globally [2,3] due partly to the increased demand for disability disaggregated data related to obligations within Article 31 of the United Nations Convention on the Rights of Persons with Disabilities [4] and to data required to report against the Sustainable Development Goals.

Fiji’s Policy on Special and Inclusive Education [5] mandates that no child will be left behind. This policy is strongly supported by the government and is actively being rolled out through an implementation plan. The existence of Special and Inclusive Education Grants for schools based on enrolment of students with disabilities requires a rigorous process for determining eligibility for the grant. An important feature in the Fijian context is the relative lack of health, rehabilitation, and diagnostic services for children with disabilities [6]. The challenge in this context is to establish a system based on robust data that draws on the resources available in the Ministry of Education—fundamentally based on teacher observations. The choice of tool to collect data from teachers is critical.

A key tool being promoted for determining disability in children to measure the disability inclusiveness of education programs in low- and middle-income countries is the UNICEF/Washington Group Child Functioning Module (CFM) [7]. This includes 24 questions regarding difficulties across 13 activity and participation domains of the International Classification of Functioning, Disability, and Health (ICF) [8]. The CFM was considered a strong candidate tool for disability disaggregation of Fiji’s EMIS (FEMIS) due to several factors. It is relatively short and simple to administer without medical expertise, it identifies children with different functional difficulties, and would enable comparison of school data to national data collections administered by the National Statistics Office (for which the CFM was designed). In addition, the use of data on functioning to identify children with disabilities is a positive shift from the diagnostic or impairment data used previously in Fiji and commonly in other low- and middle-income country EMISs [2,3].

However, validation of the CFM has shown varying accuracy (a measure of sensitivity and specificity) from “excellent” to “poor” across disability domains and only “fair” accuracy of the module as a whole [9,10,11,12]. The functional areas of seeing, hearing, walking, and speaking appear to have the greatest accuracy, whilst cognitive domains, such as learning, remembering, and focusing attention, have poorer accuracy. In addition, ”some difficulty” response category in the CFM captures a large proportion of children with no disability as well as those with moderate-severe impairments, making it difficult to estimate the level of disability among these children. Verification of disability for grant eligibility requires face-to-face school visits; however, it would be inefficient to visit all children recorded as “some difficulty”. These results highlight the need to explore the value of combining CFM data with additional information to enable a more accurate estimation of disability and reduce false positives on the list of children who need verification visits.

In the context of planning for disability-inclusive education, researchers and practitioners have long acknowledged the need to focus far more broadly than on impairments and health conditions of individual children. The focus must also be on functioning and participation considerations [13,14] and changes required in the environment [1] or instructional context [15]. To establish eligibility for special supports, information is needed that defines whether a child has a disability as well as what the child’s learning and support needs are to enable participation.

The ICF has been praised for its applicability within educational systems in classification and identifying and planning for children’s support needs [16,17,18,19]. Aljunied and Frederickson [20] observed how well-matched the ICF’s bio-psycho-social model of disability is with interactional models of assessment and ecological systems’ practice frameworks used by educational psychologists. However, despite this, there has been slow progress in embracing the ICF model in relation to children with special educational needs, perhaps due to challenges in operationalizing it [20,21,22]. Lebeer et al. [23] investigated special education needs assessments across seven European countries and found that the ICF model was used in only one country, Portugal. In other countries, static standardized psychometric testing was the prevailing method.

In ICF terms, environmental factors are facilitators or barriers influencing human functioning, which play a part in determining the extent to which health conditions lead to an experience of disability and the extent to which children with disabilities access quality education. Environmental factors are categorized as: (i) products and technology, (ii) natural environment and human-made changes to environment, (iii) support and relationships, (iv) attitudes, and (v) services, systems, and policies [8].

From her extensive examination of ICF literature [24], Madden highlighted significant challenges regarding how to incorporate a consideration of the environment and its effect on a person’s functioning and how to set thresholds to define groups of people with different levels of functioning required for population prevalence estimates and program eligibility. She noted how few existing measures embrace both functioning and environmental factors and advocated working towards a measure that incorporated activities and participation as well as environmental factors and measures of “need for support or assistance” (p. 5827).

The complexity of coding environmental factors is acknowledged in the ICF Children and Youth manual, using the example of footpath curb cuts without textured paving being a barrier for a blind person but a facilitator for a wheelchair user [25]. A student in one school who requires personal assistance for toileting may, in a different school with grab rails installed in the toilets, be independent in toileting. Another challenge relates to the dependability and variability of access to the resource, for example, access to mobility aids in a context such as Fiji are variable; a child may have a wheelchair prior to a cyclone, but then this aid is destroyed during a disaster, and replacement may not happen for several years depending on donor priorities. This highlights the challenge in using environmental factors to code disability and the absolute centrality of context.

Klein and Kraus de Camargo [19] noted the lack of an exhaustive definition of “environment” and the subjectivity of a definition of “typical environment”, proposing that environmental factors across cultural and educational contexts may place different demands on functional abilities and may therefore necessitate an adaptable definition. The ICF-based Documentation Tool [26] reflects the flexibility required for environment data by providing space for open-ended responses describing the facilitator/barrier. Regarding the environmental factor “Products and technology for education” (p. e130), the descriptor is “Equipment, products, processes, methods, and technology used for acquisition of knowledge, expertise, or skill, including those adapted or specially designed”. Similarly, the environmental factor “Special education and training services” (p. e5853) includes the descriptor “Services and programs concerned with special education in the acquisition, maintenance, and improvement of knowledge, expertise, and vocational or artistic skills, such as those provided for different levels of education, including those who provide these services”. The problem with a data-collection tool such as this in Fiji is that most teachers would be unable to describe the facilitators/barriers that relate to these descriptors. Whereas by providing a list of contextually relevant and available products and services, for example “Braille machine” or “teacher aide”, teacher respondents are more likely to be able to complete the form. Benson highlighted how “the ICF’s complexity can be bothersome” [13] (p. 10) even in well-resourced settings where the person completing the ICF form may be an educational psychologist.

In Fiji, as in most settings, the picture related to environmental factors in the education context is complicated—a new inclusive education policy with a staged implementation plan [27], changing attitudes, varying access to assistive technologies, emergent and sporadic efforts towards school accessibility modifications, and nascent availability of personal assistance in school settings are but some of the factors. To conceptualize environmental factors in a way that would be feasible to measure in the context of these dynamics and fit for purpose, we divided environmental factors into two levels: (i) the individual student and (ii) the school and broader environment, with different measurement approaches for each.

School and broader environment factors have been documented extensively [2,28,29,30] and include things such as built environment, transport, policies, flexibility of curricula, attitudes (parents, teachers, principal, peers, and community), pedagogy (teaching method and practice), teacher capacity, and economic costs. There is a degree of overlap for some of these factors between the two levels, individual student and school/broader environment, and we acknowledge that concepts such as “pedagogy” have implications that relate to both levels. This paper focuses particularly on the first level—environmental factors specific to the individual, for which we use the term “learning and support needs” (LSN).

FEMIS has a “granular” design based on individual student electronic files [3]. The implication of this system design for disability disaggregation is that information defining disability can be based on a combination of constructs from different parts of the ICF. Combining variables occurs through algorithms, which are calculations within FEMIS. An important question was whether combinations of functioning data from the CFM and additional data on environmental factors could be used to increase the accuracy of identification of children with disabilities in Fijian schools.

The objectives of this paper were to (i) determine LSN of Fijian children with different types of functional difficulties and impairments and (ii) identify combinations of functioning and LSN data (including educational adjustments, assistive technologies, and need for personal assistance) that may distinguish between disability types amongst children in Fiji.

## 2. Materials and Methods

### 2.1. Study Design and Sampling

This consisted of a nested cross-sectional survey within a larger diagnostic accuracy study [12,30] undertaken from March–July 2015 in Fiji. Ethics approvals were obtained from the University of Melbourne’s Human Research Ethics Committee and the Fiji Ministry of Education’s ethics committee. All subjects had written consent. Sampling was purposive regarding school selection and student participation. Participants for the study were 5–15-year-old students recruited from ten special schools and five inclusive education (mainstream) schools from the four administrative divisions in Fiji. Children invited to participate included: all children in the special schools and all children in the mainstream schools previously identified by the school to have disabilities and selected controls matched by age, sex, ethnicity, and location. Clinical assessments to determine impairment status are detailed elsewhere [9,10].

### 2.2. Participant Demographics

The sample included 472 children with mean ± SD age of 10.2 ± 2.6 years (range: 5 to 15) in classes 1 to 8, including approximately half from special and half from mainstream schools. Ninety-eight teachers participated. Distribution of impairments and functional difficulties have been reported elsewhere [12], but important to note for interpretation of the results is the fact that many children had multiple types of impairment, as determined by the clinical assessments.

### 2.3. Survey Tool

This study used a draft of the CFM (5–17 years age group), current at February 2015, with permission from UNICEF and the Washington Group. Translation and pretesting processes are described in earlier publications [3]. The CFM includes 24 questions covering seeing, hearing, walking, self-care, speaking, learning, remembering, concentrating, behavior, socialization, and mood [31]. Scoring essentially uses a Likert scale of severity including “no difficulty”, “some difficulty”, “a lot of difficulty”, and “cannot do at all”. UNICEF/WG advise the cut-off for counting children at risk of disability as “a lot of difficulty” or “cannot do at all” to any item. The module has two age group versions (2–4 and 5–17 years); the 5–17-year-old module was used for this study to match the primary school age. The CFM was interviewer-administered with parents/caregivers. Teachers completed the questionnaire independently, which included the CFM plus additional questions on environmental factors, that is LSN, including personal assistance, adaptations to learning or assessment, and assistive technology, as outlined below.

The LSN questions for our survey were devised based on a literature review and in collaboration with special and inclusive education experts from Fiji to ensure contextual relevance. The number of LSN questions was established to provide reasonable detail without being unwieldy, with questions worded with an average Fijian primary teacher in mind with no special education qualifications or training. The three LSN questions were:“Compared with children of the same age, how much personal assistance at school does (child’s name) require with any of the following tasks? (a) Moving around the classroom, (b) moving around outside in the school grounds, (c) getting to and from school, (d) communication, (e) cognitive/learning activities, (f) self-care (eating, toileting), (g) socializing with other children, (h) managing own behavior.” The response categories were (i) needs no extra assistance, (ii) needs a little more assistance than other children, (iii) needs much more assistance than other children.“Are there any adaptations to learning or assessment that you currently make for (child’s name)? (a) Child sits close to the board or teacher, (b) printed materials are enlarged, (c) printed materials are provided in Braille, (d) physical education (sport) activities and games are modified, (e) modifying the lesson or reducing the complexity of the lesson for the child, (f) sign language interpreters are available for learning and other school activities, (g) additional time provided for assessments, (h) personal assistance provided during assessments (e.g., note taker/writer, sign language interpreter, etc.). Response categories were (i) yes, we do this, (ii) no need for this, (iii) not done, but there might be a need.“Is (child’s name) currently using any of the following types of assistive devices? Tick all applicable options; referred to pictures of assistive devices: wheelchair, crutches, walking stick or walking frame, screen reading software, Braille machine, white cane, glasses, hearing aid, magnifier, orthotic devices, artificial limbs, modified furniture, communication boards, computer used specifically to overcome functional limitation/disability”.

### 2.4. Data Analysis

Impairment severity was determined based on clinical assessments of only vision, hearing, musculoskeletal impairment, speech, and cognition and did not cover areas such as psychosocial function, behavior, or attention. The highest level of severity in any of the five clinical assessments was taken as impairment severity. Level of functional difficulty was established by taking the highest level of difficulty in any of the CFM domains (covering a more comprehensive range of disability domains than impairment severity).

Frequencies were used to analyze relationships between assistive technology, adaptations, and assistance required and: (i) five impairments (vision, hearing, musculoskeletal, speech, and cognitive), including children with only single impairment as well as any/multiple impairments and (ii) difficulties in the functional domains not covered by the five clinical assessments (behavior, socialization, anxiety, and depression). Spearman’s Rho correlation coefficient was used to test correlation between level of assistance needed and impairment severity and level of functional difficulty. Correlation coefficients were classified as very high (0.90–1.00), high (0.70 < 0.90), moderate (0.50 < 0.70), low (0.30 < 0.50), and negligible (0.00 < 0.30) [32]. Level of assistance was cross-tabulated with impairment severity and level of functional difficulty.

## 3. Results

### 3.1. Relationship between Impairments, Assistive Technology, Adaptations, and Assistance Required

This section outlines frequencies of (i) adaptations to learning and assessment (educational adjustments) (Table 1), (ii) personal assistance required (Table 1), and (iii) use of assistive technologies for children with impairments (defined by the clinical assessments) and for children with difficulties in functional domains not covered by the clinical assessments (behavior, socialization, anxiety, and depression). The number of children with speech impairments and no other types of disability was too small to analyze these results.

As shown in Table 1, the most common educational adjustment being provided is “additional time for assessments”, provided to 91.3% of children with difficulties with behavior/attention/socialization and between 70.0–78.8% of children with other disability types except for children with only musculoskeletal impairment (MSI), of whom only 50.0% received this accommodation. The next most common LSN provided is “modifying or reducing the complexity of lessons”, followed by “student sits close to board or teacher”, “personal assistance provided during assessment”, and then “modifying PE sessions”, which was done or needed for 71.4% of children with MSI but only for 30.4% of children with difficulties with behavior/attention/socialization. Whilst “enlarged printed materials” were unsurprisingly provided to 45% of children with vision impairment, they were reportedly also used or needed for 28.9% of children with cognitive impairment (CI), 42.1% of children with hearing impairment (HI), 30.4% of children with difficulties with behavior/attention/socialization, and 38.4% of children with any/multiple disability types. Similarly, whilst “sign language interpreters” were unsurprisingly provided or needed mostly for children with HI (84.3%), they were also provided for 34.8% of children with difficulties in behavior/attention/socialization, 21.4% of children with MSI, and 33.5% of children with any/multiple disability types.

Of the LSN that require personal assistance, “assistance with cognitive/learning activities” was the most reported. Children with difficulties in behavior/attention/socialization required “much more” assistance (56.5%) than other children (and 34.8% required “a little more”) compared to only 5% and 20% of children with vision impairment needing these respective levels of assistance. About two-thirds (65.8%) of children with HI required “a little more” assistance and only 5.3% “much more”. The next most common need for personal assistance was “assistance with communication”; once again, children with difficulties in behavior/attention/socialization had greater levels of need (73.9%), followed by children with HI (73.6%), and then CI (48.1%). The third most common need was for “assistance managing own behavior”, for which 86.9% of children with difficulties in behavior/attention/socialization required either “much more” or “a little more” assistance than other children compared to 46.1% of children with CI and less for children with MSI (28.5%), HI (26.3%), and VI (10.0%). About one-third (35.7%) of children with MSI need “much more” “assistance getting to/from school” and 20–30% with “self-care” and “moving around in and outside the classroom”.

Regarding use of assistive technologies, of the children with vision impairment and no other difficulty (*n =* 20), 32.1% have printed materials provided in Braille. All six children who used Braille machines were blind; all four children who used white canes were blind; 8/12 of the children who used screen reading software had vision impairment (moderate-blind); and 10/16 of the children who used glasses had mild to severe vision impairment. All 28 of the children who used hearing aids had HI. All 14 children who used wheelchairs had MSI; all seven children with crutches/walking frame had MSI; and all four children with orthotic devices had MSI. None of the sample used prosthetics. Five of the 23 children with difficulties only in behavior/attention/socialization used a communication board; and three used a computer to support functional limitations. Of the 19 children with modified furniture, six had MSI, four were blind, and eight had CI.

The number of children who appeared anxious or depressed “daily” were too few (*n =* 10 and *n =* 5, respectively) to report frequencies usefully. However, to establish whether LSN data usefully differentiate disability domains (discussed later in relation to algorithms), it is pertinent to provide an overview of the results. None of these children had assistive technology needs. Most children with anxiety had modified lessons, additional time for assessments, and sat near the teacher or board, and small numbers needed assistance with learning and assessments, communication, behavior management, and modifying PE activities. There were no distinct patterns of LSN for children who appeared depressed “daily”. LSN varied across the children and included assistance with learning, communication, assistance socializing, managing behavior, modified lessons, additional time, personal assistance for assessments, sitting close to the board or teacher, and modified PE activities.

### 3.2. Correlation and Cross-Tabulation between LSN, Impairment Severity, and CFM Responses

Overall, there was a significant ‘‘moderate’’ (r = 0.519; *n =* 390; *p* < 0.000) correlation between the level of assistance needed and impairment severity (based on five impairment types). There was a significant “moderate” (almost “high”) correlation (r = 0.681; *n =* 390; *p* < 0.000) between the level of assistance needed and level of functional difficulty (based on teacher responses to all CFM questions).

The left-hand side of Table 2 presents the spread of level of assistance required across the levels of functional difficulty reported in the CFM. As expected, the level of assistance required increases proportionally with the level of functional difficulty. Of children with the level of functional difficulty at “a lot of difficulty” and “cannot do at all”, 44.2% and 76.1%,, respectively needed much more assistance than other children. Of the children with “some difficulty” functioning, 38.5% needed no assistance, and 45.3% needed only a little more assistance than other children; and of the children with “no difficulty” functioning, 89.6% required no assistance.

Of the 16.2% of children reported as having only “some difficulty” and yet who needed much more assistance than other children (*n =* 19), 84.2% had impairments; 14 were severe and two moderate. Of the children with “some difficulty” functioning who needed “a little more assistance” (*n =* 53), 47.2% had impairments; 14 were severe, 10 moderate, and one mild. Of those with “some difficulty” yet requiring “no assistance” (*n =* 45), only 22.2% had impairments; three severe, six moderate, and one mild.

The right-hand side of Table 2 presents the spread of level of assistance required across the impairment severities. Children with severe impairments had the highest assistance needs, with 62.5% requiring much more assistance and 46.0% requiring a little more assistance than other children. Most children with no impairments required either no assistance (58.3%) or only a little more assistance (27.4%) than other children. As mentioned earlier, the impairment severity only considers assessments of vision, hearing, musculoskeletal, speech, and cognition. The raw data were reviewed to explore factors that may explain the 24 children who appear to have “no impairment” but require “much more assistance” than other children; all 24 children fit into one or more of the following categories: anxious “daily”, depressed “daily”, have more or a lot more difficulty controlling their behavior, have a lot of difficulty accepting changes to their routine, have a lot of difficulty making friends, or are reported as having particular learning difficulties, such as dyscalculia.

## 4. Discussion

FEMIS requires a higher degree of accuracy than most EMISs in low- and middle-income countries because it is the basis of funding eligibility decisions at an individual student level. Within an overarching goal of developing a valid and feasible approach for disability disaggregation of FEMIS, previous work [12] established that functioning data from the CFM are not accurate enough to identify disability for this purpose. In addition, as FEMIS is also used to document required educational accommodations for individual children, functioning data do not provide this information. Due to the granular nature of FEMIS’s architecture, multiple variables can be combined within algorithms (calculations) in FEMIS to define disability types and levels. This study explored LSN of Fijian children with different types of functional difficulties and impairments and explored combinations of functioning and LSN data that may distinguish between disability types amongst children in Fiji.

In Fiji, disability-inclusive education is a relatively emergent approach. It is early days in building teacher skills in differentiating teaching for children, discerning different LSN, and awareness of options for reasonable accommodations. In addition, resources to thoroughly regulate the new Special and Inclusive Education Policy are not available currently. In this context, it is likely that using LSN data as the primary means of determining eligibility for disability funding would have lower validity and reliability than observations of functioning.

As outlined earlier, classification merely by diagnosis does not adequately inform supports needed for individual children [19], and diagnosis has been shown to be a weak predictor of participation compared with environmental factors [29,33]. Especially evident in diagnoses, such as autism spectrum disorder, learning disorders, or cerebral palsy, there is enormous variation in functional abilities within and across these diagnoses. Ruppar et al. [15] argued that assigning resources in education systems should not happen based on disability labels but instead by careful consideration of LSN.

### 4.1. Learning Support Needs

The most common LSN identified in our study were requirements for additional time and personal assistance during assessments, modifying or reducing the complexity of lessons, providing personal assistance with cognitive/learning activities, and sitting close to the board or teacher. These needs were mutual to children with all types of impairments or functional difficulties. The widespread use of these cost-free and allowable modifications by teachers in the study sample is a positive indication of knowledge and application of educational accommodations for students with disabilities. Assistance with communication and managing behavior were also commonly required, highlighting the need for teacher skills in positive behavior management and methods for building communication skills in children.

A small number of LSN were specific to impairment groups, for example, providing materials in Braille was only relevant for children with vision impairment, and assistance moving around the classroom or outside was mainly relevant for children with mobility impairment. Other LSN were predominant amongst certain types of impairments or functional difficulties but were required across a wide range of children, for example, modifying PE activities and enlarging printed materials. Sign language interpreters were used mostly for children with HI but also for children with speech and cognitive impairments.

Results regarding the need for personal assistance inform human resource planning for disability-inclusive education. Children with difficulties with behavior/attention/socialization needed high levels of personal assistance across a range of tasks. On the other hand, children with vision impairment required the least personal assistance, although the sample are almost entirely from a specialist school for children with vision impairment. It is possible that in a well-adapted environment where educational accommodations are in place, students with vision impairment are relatively independent of additional personal supports.

### 4.2. Combining LSN and Functioning Data

The sensitivity of the CFM response category “a lot of difficulty” was too low, and the sensitivity of the response category “some difficulty” was high, but the specificity was very low. This implies that many children on the Ministry of Education’s list for conducting disability verification visits to schools (based on the response option “some difficulty”) would be found not to have disability, wasting resources for unnecessary visits. Our findings suggest that combining LSN data with CFM data could address this challenge with the “some difficulty” category.

The LSN data effectively distinguished between children with and without disabilities, and the needs increased proportionately along a gradient of increasing severity of impairment and functional difficulty. Amongst children reported as having “some difficulty”, the gradient of impairments directly related to the reported levels of assistance required. The implication of this is that combining functioning data from the CFM with LSN data may increase the accuracy in identifying children with disabilities amongst those identified as only having “some difficulty” on the CFM.

The results showed that some assistive technologies were useful for distinguishing between disability types and can be useful for this purpose in algorithms combining CFM data with LSN; these include hearing aids, Braille machine, white cane, wheelchair, orthotic devices, and prosthetics. On the other hand, four assistive technologies were used by children across a range of disability types and were therefore unable to be used within algorithms in the same way. These include modified furniture, screen reading software, communication board, and computer used to support functional limitation. Collecting information on these assistive technologies may be useful to determine the LSN of an individual child, but as elements in algorithms designed to delineate disability types, they are confounding variables.

The intersection between students’ capacities and environmental factors, or LSN, is the basis to defining supports and services needed for successful educational outcomes of students with disabilities [34]. Various researchers have underscored the value of the ICF for considering and documenting the role of the environment as a barrier or facilitator of child functioning, including accommodations and pedagogical modifications [14,17], although widespread uptake of the ICF for this purpose has been limited [19]. A common tool for disability-inclusive education in many countries, facilitating communication and cohesive approaches between teachers, families, and others involved in the education of children with disabilities is the Individualized Education Plan (IEP). Central to the development of the IEP is identifying a child’s LSN and establishing an agreed process of meeting these [35]. Whilst the exhaustiveness of the ICF-CY coding has been criticized as being unwieldy [13], Kostanjsek et al. [36] envisaged the possibility of developing a list of generic environmental factors relevant across health conditions that could be implemented alongside functioning questions. This is the approach taken within Fiji.

### 4.3. Student Learning Profile

Based on the study results, Fiji developed an assessment tool called the Student Learning Profile. This incorporates functioning data based on the CFM, a generic list of questions on environmental factors/LSN including assistive technology, and student strengths and interests. The Student Learning Profile is the basis of algorithms in FEMIS that combine functioning and LSN data to define disability and distinguish between disability types. It is the basis of school discussions to develop a student’s IEP. Sanches-Ferreira et al. [14] emphasized the importance of severity of functional difficulty for determining eligibility and planning for appropriate supports; accordingly, the Student Learning Profile includes four CFM response categories to inform severity.

Data from CFM is the most useful element in distinguishing between disability types within the algorithms, whereas most LSN questions do not help in distinguishing between disability types because they are applicable across a wide variety of children. However, when used in combination with CFM functioning data, the LSN are very helpful in distinguishing between children with and without disabilities, which is vital for accuracy of disability identification amongst children who are reported as having “some difficulty” on the CFM.

### 4.4. International Comparability-Environmental Factors Versus Functioning Data

Mont pointed out that environmental factors, such as educational accommodations, vary across contexts and over time and are heavily dependent on policies and resourcing and that therefore in establishing an internationally comparable method for disaggregating EMISs, using learning supports to categorize disability is inappropriate; in contrast, functioning data are more suited for this purpose [37]. Hollenweger [1] agreed that environmental factors vary greatly across contexts but argued that “ultimately, the policy context, financial resources, and available services define which eligibility criteria are applied and how they are applied” (p4) and that it is therefore doubtful that a uniform disability definition that ignores contextual influences would result in equitable supports for inclusive education. In terms of FEMIS, using functioning and environmental factors data to categorize disability for disaggregating FEMIS enables the provision of equitable supports, and where needed (for internationally comparable data), the functioning data could be extracted from the system separately from the LSN data.

### 4.5. Limitations

A limitation of this study is the inability for the findings to be transferred to some low- and middle-income contexts at this point. As highlighted earlier, the granular nature of FEMIS enables disability determination based on combinations of functioning and LSN data using algorithms. This is not possible in countries using a manual, census-based EMIS. However, as shown in UNESCO’s recent status assessment of EMISs globally, an increasing number of EMISs globally use (54%) or are transitioning to (34%) granular forms [3,38].

A further limitation relates to the lack of attention paid in this paper to environmental factors at the school and broader environment level. Factors, such as accessible buildings and assessment policies, play a central role in the degree to which a child with a health condition is disabled. Whilst the categorization of disability for the purpose of disaggregating FEMIS is based on individual student level factors identified in the Student Learning Profile, data on school and broader environment level factors is collected within FEMIS using a School Accessibility and Inclusion Form [39]. Data from this form are entered on each the school’s page within FEMIS (as opposed to the individual student pages). This enables analyses, such as correlations between individual student learning outcomes, with school and broader environment level factors.

## 5. Conclusions

The method of disaggregating FEMIS by disability requires more accuracy than most EMISs in low- and middle-income countries because it is the basis of funding eligibility decisions at an individual student level. We have shown that combining activity and participation data from the CFM with data on environmental factors (i.e., learning and support needs) through algorithms enables domain-specific disability identification for the purpose of disability disaggregating FEMIS. Certain LSN are common to children with various disability types and, whilst useful in identifying which children have disability amongst those reported as having “some difficulty” on the CFM, these LSN items are not as useful in distinguishing between disability domains. CFM data are more important in distinguishing between disability domains. Additionally, data on LSN support teachers to develop and monitor IEP for students with disability.

Fiji’s Policy on Special and Inclusive Education [5] includes measures for providing schools with funding through the Special and Inclusive Education Grant. The data used to determine student eligibility are derived from the disability algorithm that resulted from this study, programmed in FEMIS. Students identified through the algorithm are verified by district education officers. The grant is used to fund equipment, aids and resources, specialized supports, school infrastructure and transport accessibility measures, accessibility of sport and recreational activities, and costs related to referral services.

## Figures and Tables

**Table 1 ijerph-18-09413-t001:** Learning and support needs as a percentage of children with disability, including any/multiple disability types and children with single-disability categories.

Learning and Support Need		Any/Multiple Disab. (*n =* 245)	VI (*n =* 20)	HI (*n =* 38)	MSI (*n =* 14)	CI (*n =* 52)	Beh/Att/Soc (*n =* 23)
Educational Adjustments currently provided for learning and assessment							
Additional time provided for assessments	Y	78.8	70	71.1	50	75.5	91.3
	NbN	4.5	0	13.2	7.1	1.9	4.3
	Total	83.3	70	84.3	57.1	77.4	95.6
Student sits close to board or teacher	Y	63.3	45	73.7	50	57.7	78.3
	NbN	3.3	0	2.6	0	1.9	0
	Total	66.6	45	76.3	50	59.6	78.3
Lessons modified or reduced in complexity	Y	73.5	55	68.4	57.1	67.3	78.3
	NbN	2.4	0	5.3	0	3.8	4.3
	Total	75.9	55	73.7	57.1	71.1	82.6
Personal assistance provided during assessment	Y	62.9	40	76.3	35.7	50	69.6
	NbN	6.1	10	2.6	7.1	7.7	4.3
	Total	69	50	78.9	42.8	57.7	73.9
PE sessions are modified	Y	64.5	50	55.3	64.3	50	30.4
	NbN	4.5	0	5.3	7.1	1.9	0
	Total	69	50	60.6	71.4	51.9	30.4
Enlarged printed materials provided	Y	33.9	45	39.5	21.4	23.1	30.4
	NbN	4.5	0	2.6	0	5.8	0
	Total	38.4	45	42.1	21.4	28.9	30.4
Sign language interpreters used	Y	33.5	10	79	21.4	11.5	34.8
	NbN	5.7	0	5.3	7.1	1.9	8.7
	Total	39.2	10	84.3	28.5	13.4	43.5
Personal Assistance required for tasks							
Needs assistance with cognitive/learning activities	Little	43.3	20	65.8	28.6	50	34.8
	Much	28.2	5	5.3	14.3	21.2	56.5
	Total	71.5	25	71.1	42.9	71.2	91.3
Needs assistance with communication	Little	35.1	20	44.7	21.4	42.3	34.8
	Much	22.9	0	28.9	7.1	5.8	39.1
	Total	58	20	73.6	28.5	48.1	73.9
Needs assistance managing own behavior	Little	35.5	5	26.3	21.4	42.3	47.8
	Much	17.1	5	0	7.1	3.8	39.1
	Total	52.6	10	26.3	28.5	46.1	86.9
Needs assistance getting to/from school	Little	24.1	10	44.7	14.3	11.5	26.1
	Much	15.5	10	5.3	35.7	1.9	13
	Total	39.6	20	50	50	13.4	39.1
Needs assistance socializing with other children	Little	22	5	13.2	21.4	21.2	21.7
	Much	9	0	0	0	0	26.1
	Total	31	5	13.2	21.4	21.2	47.8
Needs assistance with self-care	Little	19.2	0	10.5	21.4	11.5	34.8
	Much	12.7	0	0	21.4	3.8	13
	Total	31.9	0	10.5	42.8	15.3	47.8
Needs assistance moving around in classroom	Little	9	10	10.5	21.4	3.8	4.3
	Much	7.3	5	0	21.4	0	0
	Total	16.3	15	10.5	42.8	3.8	4.3
Needs assistance moving around outside	Little	9.4	5	5.3	21.4	3.8	8.7
	Much	9.4	10	2.6	28.6	0	0
	Total	18.8	15	7.9	50	3.8	8.7

Y, “Yes, we do this”; NbN, “No, but there might be a need”; Little, “Child needs a little more assistance than other children”; Much, “Child needs much more assistance than other children”. Classified by clinical assessments: VI, vision impairment (visual acuity < 6/18) [8]; HI, hearing impairment (≥41 dBA) [8]; MSI, musculoskeletal impairment (classified as moderate-severe musculoskeletal impairment on the Rapid Assessment of Musculoskeletal Impairment) [8]; CI, cognitive impairment [10]. Classified by teacher CFM score of ≥ “A lot of difficulty” on one or more of the respective questions: Behavior/Attention/Socialization.

**Table 2 ijerph-18-09413-t002:** Cross-tabulations between level of assistance required and highest level of functional difficulty (CFM response) and impairment severity.

	Highest Level of Functional Difficulty *n* (%)				Impairment Severity *n* (%)			
	No Difficulty	Some Difficulty	A Lot of Difficulty	Cannot Do at all	None	Mild	Moderate	Severe
No assistance needed	69 (55.2)	45 (36.0)	9 (7.2)	2 (1.6)	98 (78.4)	6 (4.8)	14 (11.2)	7 (5.6)
Needs a little more assistance than other children	7 (5.1)	53 (38.7)	63 (46.0)	14 (10.2)	46 (33.6)	5 (3.6)	23 (16.8)	63 (46.0)
Needs much more assistance than other children	1 (0.8)	19 (14.8)	57 (44.5)	51 (39.8)	24 (18.8)	6 (4.7)	18 (14.1)	80 (62.5)
Total	77 (19.7)	117 (30.0)	129 (33.1)	67 (17.2)	168 (43.1)	17 (4.4)	55 (14.1)	150 (38.5)

## Data Availability

The data presented in this study are available on request from the corresponding author. The data are not publicly available due to the ethical protocols and to ensure privacy of participants.

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
