# Peer review of "Combining Child Functioning Data with Learning and Support Needs Data to Create Disability-Identification Algorithms in Fiji’s Education Management Information System"

_ijerph, 2021, doi:10.3390/ijerph18179413_

Round 1
Reviewer 1 Report
RESUME
The study objective is well defined and identified both in the abstract and in the introduction.
INTRODUCTION:
The subject investigated is of scientific and social interest.
The authors point out the difficulty of defining disability and argue the choice of the ICF to do so, supporting it with the necessary bibliography.
The authors comprehensively identify the problems that can be found in applying this definition of disability to children and how to solve them, helping to clarify ideas.
The ideas presented are well argued with the bibliography that they detail.
MATERIALS AND METHODS - DISCUSSION AND RESULTS
The statistical apparatus is well treated and the results are well analyzed.
CONCLUSION
I would like to see some more conclusion, in relation to the action of public policies, that would be an orientation to achieve a more effective equality of opportunities.
Author Response
Thank you for the positive feedback. We have included information on how the Special and Inclusive Education Policy includes the mandate for students with disabilities to receive the Special and Inclusive Education Grant, based on data derived from the algorithm in FEMIS.
Reviewer 2 Report
I appreciate the opportunity to analyze this article, which for the reasons I will explain, I think is very important. Below are some suggestions that may lead to improvements in the text:
(a) The identification of educational needs in this population must go beyond what is supposed in the international classification. In order to build a more humane system, there is a set of dimensions that must be considered and operationalized: Learning difficulties, Communication problems, Mental disability, Behavior disorders, Multi-disability, Hearing impairment, Motor problems, Other Health Problems: tuberculosis, epilepsy , hemophilia, asthma, … Attention deficit hyperactivity disorder: lack of
attention, hyperactivity, impulsivity, visual impairment, autism: Asperger syndrome, deaf blindness, educational risk and giftedness”. It is important to consider these aspects in order to become more humanized and supportive in the qualification of the human and material resources to be mobilized.
(b) Please indicate the reference (n.) of the Ethics Committee
(c) LSN questions - Some dimensions I mentioned earlier are missing. However, as the tool is built, maybe include some variables in the first part, referring to the sociodemographic and clinical characterization;
(d) The Student Learning Profile should include other variables for its operationalization, but I give the authors the benefit of the doubt because I don't know the cultural context. However, I must add that my evaluation of this study is very positive, as the humanization of education, the preparation of children who have the right to intervene in the social context and even Human Rights are at stake.

Author Response
(a) The identification of educational needs in this population must go beyond what is supposed in the international classification. In order to build a more humane system, there is a set of dimensions that must be considered and operationalized: Learning difficulties, Communication problems, Mental disability, Behavior disorders, Multi-disability, Hearing impairment, Motor problems, Other Health Problems: tuberculosis, epilepsy , hemophilia, asthma, … Attention deficit hyperactivity disorder: lack of attention, hyperactivity, impulsivity, visual impairment, autism: Asperger syndrome, deaf blindness, educational risk and giftedness”. It is important to consider these aspects in order to become more humanized and supportive in the qualification of the human and material resources to be mobilized.
- Response: Thank you. No doubt the specific health conditions a child may have can be very important in determining how teachers should craft their individual education plans. In Fiji unfortunately there are fundamentally no diagnostic services available (aside from occasional visiting experts and volunteers), and so the context of the research is to build upon the functional observation skills that teachers have. In the introduction section, we frame the paper by acknowledging the limitations and critique made by researchers and practitioners of reliance on diagnostic categories when understanding functional capacity and education support needs.
(b) Please indicate the reference (n.) of the Ethics Committee
- Response: Thank you. The reference number of both the University of Melbourne and Fiji’s Ministry of Education, Heritage and Arts are included in the Institution Review Board Statement at the end of the manuscript.
(c) LSN questions - Some dimensions I mentioned earlier are missing. However, as the tool is built, maybe include some variables in the first part, referring to the sociodemographic and clinical characterization;
- Response: Thank you. The student’s file in FEMIS includes sociodemographic information, and space has been built into the system for parents to record medical reports if they have them.
(d) The Student Learning Profile should include other variables for its operationalization, but I give the authors the benefit of the doubt because I don't know the cultural context. However, I must add that my evaluation of this study is very positive, as the humanization of education, the preparation of children who have the right to intervene in the social context and even Human Rights are at stake.
- Response: Thank you so much for your encouragement. We have worked hard to develop a system that contributes to the achievement of the rights of children with disabilities.
Reviewer 3 Report
The present work makes an interesting contribution in the line of determining with greater precision certain types of disability considering multiple factors. It presents an adequate structure and arrangement of the data, a correct methodological approach and a good presentation of the results and conclusions.
Author Response
Thank you for your positive feedback.
Reviewer 4 Report
Here, the authors describe a cross-sectional survey of students with disabilities and their teachers in Fiji, with a goal to assess the distribution of levels of disability and learning and support needs. Among 472 children and 98 teachers, the authors found much expected relationships between learning/support needs and some disabilities like hearing and vision impairments, while others were more spread across different disability types. The authors conclude that while the existing Child Function Module is able to distinguish most disability types, the addition of learning and support needs parameters may help to better identify lower disability-level as having disability in need of intervention from those with negligible disability.
This is an area of public health research which I am not particularly familiar with. Even still I will make some fairly general comments. It would be useful to have someone with more expertise in learning and particularly children with disabilities review this article as they would have greater appreciation of the methods and approaches in this area of research.
- In the Abstract, the Background says the CFM alone is not accurate enough to identify disabilities for the purposes of disability disaggregation but then the Results says the CFM is more useful than LSN data in identifying disability domains. This seems a conflict. While the end conclusion is that LSN in addition to CFM would be a more robust mode of assessing disability amongst children, particularly those of lower disability level, I might suggest some rephrasing of the Background, perhaps to be more conservative and say the CFM may not be accurate enough to identify all disability levels?
- On line 72, I might suggest rephrasing this sentence to say “disability level among these students cannot be known”, rather than suggesting some students would be ignored.
- Minor quibble but I might reduce the abbreviations as there are quite a number. For instance, CRPD is used just twice so maybe just spell it out and don’t abbreviate.
- It may be a difference in discipline but I found the Introduction to be exceedingly long. I feel it could be shortened somewhat. It is also a bit out of order, discussing disability for a page and then defining disability, for instance. I might thus suggest shortening this section.
- In Table 1, I presume the cells with dashes are 0? If so, perhaps just put 0.
- In Table 2, how is it possible that 2 of the students whose level of disability was “Cannot do at all” also required no assistance?
- As with the Introduction, I found the Discussion to be quite long. I believe this could be shortened somewhat.
Author Response
- In the Abstract, the Background says the CFM alone is not accurate enough to identify disabilities for the purposes of disability disaggregation but then the Results says the CFM is more useful than LSN data in identifying disability domains. This seems a conflict. While the end conclusion is that LSN in addition to CFM would be a more robust mode of assessing disability amongst children, particularly those of lower disability level, I might suggest some rephrasing of the Background, perhaps to be more conservative and say the CFM may not be accurate enough to identify all disability levels?
Response: Thank you for your review. The results indicate that CFM alone is not accurate enough to identify disabilities but combining it with LSN data will have better accuracy. The abstract is now revised to clarify this.
- On line 72, I might suggest rephrasing this sentence to say “disability level among these students cannot be known”, rather than suggesting some students would be ignored.
Response: This is edited
- Minor quibble but I might reduce the abbreviations as there are quite a number. For instance, CRPD is used just twice so maybe just spell it out and don’t abbreviate.
Response: This is edited.
- It may be a difference in discipline, but I found the Introduction to be exceedingly long. I feel it could be shortened somewhat. It is also a bit out of order, discussing disability for a page and then defining disability, for instance. I might thus suggest shortening this section.
Response: We revised the introduction section to reduce the length.
- In Table 1, I presume the cells with dashes are 0? If so, perhaps just put 0.
Response: Yes, the dashes should be zeros and they are revised now.
- In Table 2, how is it possible that 2 of the students whose level of disability was “Cannot do at all” also required no assistance?
Response: two of the blind students in the school for the blind are fully independent.
- As with the Introduction, I found the Discussion to be quite long. I believe this could be shortened somewhat.
Response: Thank you. We’ve shortened this to increase readability.
Reviewer 5 Report
Your study explored whether combining activity and participation data from the CFM with data on environmental factors specific to learning and support needs (LSN) more accurately identifies children with disabilities. It is an interesting topic.
In my opinion, some problem could be rewrited.
1) In Section Introduction,there are so many paragrahs that I cannot accurately understand your phenomenon, issues, and a conclusion. What's important, I also don't find the marginal contribution and research in your study.
2) In Section Discuss, there are also many paragrahs and you need to increase the section headings to discuss the results of the study.
Author Response
Thank you, we revised the introduction and discussion sections to reduce the length and included section headings for better readability.
Round 2
Reviewer 5 Report
Thank you for revising the Section Introduction and Section Discussion.